# Peptidoglycan-Free Bacterial Ghosts Confer Enhanced Protection against *Yersinia pestis* Infection

**DOI:** 10.3390/vaccines10010051

**Published:** 2021-12-30

**Authors:** Svetlana V. Dentovskaya, Anastasia S. Vagaiskaya, Mikhail E. Platonov, Alexandra S. Trunyakova, Sergei A. Kotov, Ekaterina A. Krasil’nikova, Galina M. Titareva, Elizaveta M. Mazurina, Tat’yana V. Gapel’chenkova, Rima Z. Shaikhutdinova, Sergei A. Ivanov, Tat’yana I. Kombarova, Vladimir N. Gerasimov, Vladimir N. Uversky, Andrey P. Anisimov

**Affiliations:** 1Laboratory for Plague Microbiology, Especially Dangerous Infections Department, State Research Center for Applied Microbiology and Biotechnology, 142279 Obolensk, Russia; dentovskaya@obolensk.org (S.V.D.); vagaiskaya@obolensk.org (A.S.V.); platonov@obolensk.org (M.E.P.); sasha_trunyakova@mail.ru (A.S.T.); ek.al.krasilnikova@gmail.com (E.A.K.); titarevag@mail.ru (G.M.T.); elizavetamazurina99@yandex.ru (E.M.M.); tgapelchenkova@mail.ru (T.V.G.); shaikhutdinova@yandex.ru (R.Z.S.); ivanov@obolensk.org (S.A.I.); 2Department of Disinfectology, State Research Center for Applied Microbiology and Biotechnology, 142279 Obolensk, Russia; Sa777@mail.ru (S.A.K.); ilcvngerasimov@obolensk.org (V.N.G.); 3Laboratory of Biomodels, State Research Center for Applied Microbiology and Biotechnology, 142279 Obolensk, Russia; kombarova.tatyana@yandex.ru; 4Department of Molecular Medicine and Byrd Alzheimer’s Research Institute, Morsani College of Medicine, University of South Florida, Tampa, FL 33612, USA; vuversky@usf.edu

**Keywords:** *Yersinia pestis*, vaccine, guinea pigs, bubonic plague, inactivated vaccine, phage, bacterial ghost, protection, protein-E-mediated lysis, holin-endolysin system

## Abstract

To develop a modern plague vaccine, we used hypo-endotoxic *Yersinia* *pestis* bacterial ghosts (BGs) with combinations of genes encoding the bacteriophage ɸX174 lysis-mediating protein E and/or holin-endolysin systems from λ or L-413C phages. Expression of the protein E gene resulted in the BGs that retained the shape of the original bacterium. Co-expression of this gene with genes coding for holin-endolysin system of the phage L-413C caused formation of structures resembling collapsed sacs. Such structures, which have lost their rigidity, were also formed as a result of the expression of only the L-413C holin-endolysin genes. A similar holin-endolysin system from phage λ containing mutated holin gene *S* and intact genes *R-Rz* coding for the endolysins caused generation of mixtures of BGs that had (i) practically preserved and (ii) completely lost their original rigidity. The addition of protein E to the work of this system shifted the equilibrium in the mixture towards the collapsed sacs. The collapse of the structure of BGs can be explained by endolysis of peptidoglycan sacculi. Immunizations of laboratory animals with the variants of BGs followed by infection with a wild-type *Y. pestis* strain showed that bacterial envelopes protected only cavies. BGs with maximally hydrolyzed peptidoglycan had a greater protectivity compared to BGs with a preserved peptidoglycan skeleton.

## 1. Introduction

Plague, an infamous zoonotic bacterial infection caused by *Yersinia pestis*, has claimed more than 200 million human lives. For just over 100 years since the discovery of its etiological agent, several attempts have been made to develop a plague vaccine. The killed and live plague vaccines of the first generation were successfully used to save tens of millions of people, but they had a number of shortcomings. For example, live plague vaccines in pregnant women pose a theoretical risk to the fetus [1]; killed vaccines are ineffective against pneumonic plague [2]; and subunit vaccines containing only one or two immunodominant antigens (F1 and/or V) do not protect against infection with strains lacking F1 and/or producing poorly cross-reacting atypical V antigen clades [3,4]. In addition, being highly protective for mice infected with *Y. pestis* typical strains, they are less protective for similarly challenged guinea pigs and monkeys [5]. In turn, the water-insoluble “residual” antigen from yersiniae bacterial cell walls initiating a classical T cell-modulated state of cellular immunity reliably protects guinea pigs and monkeys, but is ineffective for mice [5,6,7,8].

A World Health Organization (WHO) Plague vaccine Target Product Profile (TPP) was developed to provide recommendations to vaccine developers (Available online: https://cdn.who.int/media/docs/default-source/blue-print/plaguevxeval-finalmeetingreport.pdf?sfvrsn=c251bd35_2 (accessed on 29 December 2021)). Previous immunization trials of the current vaccine candidates showed promising potential in inducing protective immunity [9,10,11,12,13,14]. However, a generally recognized plague vaccine has not yet been licensed, since previous trials did not show full compliance of candidate vaccines with modern WHO requirements.

Bacterial ghosts (BGs) are intact cytoplasm-and-its-constituents-free cell walls of Gram-negative bacteria retaining a three-dimensional cellular structure of the original microbe and generated by bacteriophage ɸX174 gene *E*-mediated lysis or other gentle biological or chemical poring methods [15]. Developed as alternative-killed vaccines with improved safety, BGs act as efficient carriers that enhance the immunogenicity of protein subunits and can express multiple intact antigens. To date, such bacteria as *Salmonella enteritidis* [16], *Vibrio parahaemolyticus* [17], *Pasteurella multocida* [18], *Bordetella bronchiseptica* [19], and *Yersinia enterocolitica* [20] have been successfully used to produce BGs. To estimate the protective potency of the *Y. pestis* ghost vaccine, we exploited individual and various combinations of the local E-mediated peptidoglycan hydrolysis with complete murein-hydrolyzing ability of the λ [21] or L-413C [22] phage holin-endolysin systems to produce BGs. In this report, we evaluated novel ghost plasmids encoding single protein E, both pairs of holin-endolysin from the both phages separately, and protein E with the each holin-endolysin pair in tandem. The purpose of the design of different constructs was to generate diverse degrees of lysis for evaluation of the interrelation of the degree of degradation of the bacterial cell wall with the intensity of immunity to plague infection in mice and guinea pigs.

## 2. Materials and Methods

### 2.1. Bacterial Strains, Plasmids, and Culture Conditions

All bacterial strains and plasmids used in this study are listed in Table 1. The *Escherichia coli* strain was grown routinely at 37 °C in Luria–Bertani broth (LB) or LB solidified with 1.2% Bacto Agar (Difco Michigan, Detroit, MI, USA). *Y. pestis* bacteria were grown routinely at 28 °C in brain heart infusion (BHI; HiMedia Laboratories, Mumbai, India) broth. Ampicillin (100 µg/mL, Amp), chloramphenicol (20 µg/mL, Cm) or 5% sucrose were supplemented as appropriate.

### 2.2. Animals

Seven-week-old male and female BALB/c mice (Lab Animals Breeding Center, Shemyakin and Ovchinnikov Institute of Bioorganic Chemistry, Pushchino, Russia) and four-week-old guinea pigs of both sexes (Lab Animals Breeding Center, Russian Academy of Medical Sciences, Stolbovaya, Moscow Region, Russia) were housed in polycarbonate cages and maintained in a light-controlled (lights on from 7:00 a.m. to 7:00 p.m.) BSL3 room at the State Research Center for Applied Microbiology and Biotechnology. The temperature and the humidity of the animal room were maintained at 22 °C ± 2 °C and 50% ± 10%, respectively. Rodents were given tap water and mouse mixed fodder PK-120 or rabbit/guinea pig mixed fodder PK-122 (Laboratorkorm, Moscow, Russia) ad libitum throughout the study. The number of animals used for the experiments were kept at the minimum dictated by the necessity, which was determined from the power analysis. The herds were divided into all groups randomly. In this study, we have used humane endpoints for the infected animals. According to the animal protocol, the mice and guinea pigs should be euthanized in the animal survival studies when they became either of the following: lethargic, dehydrated, moribund, unable to rise, non-responsive to touch, or lost more than 10% body mass.

Humane euthanasia using compressed CO_2_ gas followed by cervical dislocation has been used by well-trained individuals. We monitored the health condition of the animals at least twice a day. There were no unexpected deaths during the entire set of experiments.

### 2.3. Construction of Hypo-Endotoxic Y. pestis Strain

Construction of *Y. pestis* Δ*lpxM* derivate of the avirulent KM260(12) strain was performed as described in our previous report [25].

### 2.4. Generation of BGs

All primers used in this study are listed in Table 2. For vector plasmid construction, fragment containing p15A *ori* and *cat* gene was amplified using primers pEY-1 and pEY-2. pR’ bacteriophage λ promoter was amplified using primers pR-For and pR (T > C). Multiple cloning site from pET32b (+) plasmid and *rrnB* transcription terminator from pBAD/myc-HisA plasmid was amplified using primers pEY-5–pEY-6 and pEY-7–pEY-8 respectively and was joined by splicing overlap extension PCR with primers pEY-5 and pEY-8. After that all fragments were digested with appropriate restriction enzymes and ligated. Resulted construction pEYR’ (Figure 1) was transformed into *E. coli* DH5α and sequenced.

For pEYR’-E and pEYR’-Y-K plasmids, lysis genes were cloned into NdeI-XhoI cloning sites. For pEYR’-S-R-Rz, pEYR’-E-Y-K and pEYR’-E-S-R-Rz S-R-Rz and Y-K plasmids, genes with their own ribosome binding sites were cloned into SalI-XhoI cloning sites. The *E* gene was cloned into NdeI-SalI cloning sites.

The resulting constructs were transformed into *Y. pestis* KM260(12)Δ*lpxM* for *Y. pestis* bacterial ghost (YP-BGs) generation and cultured overnight at 28 °C. Plasmid pEYR’ lacking the lysis genes was used in control experiments. The induction of lysis was achieved by shifting the temperature from 28 °C to 42 °C when the cultures was grown to OD_600_ = 0.6, and the procedure was monitored by the optical densities. Lysed cells were harvested 24 h after the induction by centrifugation at 5000× *g* for 15 min, followed by three times washing in sterile distilled water containing 0.1 mg/mL streptomycin (in order to kill the surviving bacteria), and finally lyophilized. Freeze-dried BGs were used in subsequent electron microscopy and experiments on the immunization of animals. The resulting ghosts were named E-BG, EYK-BG, YK-BG, SRRz-BG, and ESRRz-BG, and the lysis rate was detected by counting CFUs, as described previously [26].

### 2.5. Transmission Electron Microscopy (TEM)

Sample preparation for TEM (FEI Tecnai G2 Spirit BioTWIN, Brno, Czech Republic) was carried out by centrifugation of the freeze-dried BGs, resuspended in distilled water, at 3000× *g* for 10 min, before fixing with 2.5% glutaraldehyde (pH 7.2) for 2–3 h at 4 °C. Samples were washed three times for 10 min with a 4% solution of glutaraldehyde in 0.2 M Na-cacodylate buffer, pH 7.2. Fixation was carried out overnight at 4 °C. Additional fixation of bacteria was carried out in a 4% aqueous solution of osmium tetroxide in a Reiter-Kelenberger buffer overnight at 4 °C. After fixation and washing in the buffer, bacterial samples were dehydrated for 10 min in solutions of ethyl alcohol of increasing concentration (30%, 50%, 70%, 95%) and for 20 min in absolute alcohol with its threefold change. Next, the samples were impregnated with mixtures of absolute ethanol and araldite (ratio 3:1; 1:1; 1:3) at 37 °C for a day, transferred into pure araldite, and kept in a vacuum (10^−2^ torr) for 1.5 h at a temperature of 37 °C. The samples were poured with araldite and polymerized at a temperature of 40 °C overnight, then at a temperature of 60 °C for one day, at a temperature of 90 °C for two days. Sections of fixed bacteria were obtained with a glass knife on an Ultracut ultramicrotome (Reichert Jung, Austria). The sections were contrasted with uranyl acetate and lead citrate and viewed at an accelerating voltage of 120 kV and magnification from 10,000 to 100,000 times. Electron microscopic images were taken using a high-contrast wide-angle high-resolution CCD camera GatanOriusSC200W 120 kV, as well as a high-resolution CCD camera, the GatanOriusSC 1000 V 200 kV. 

### 2.6. Ethics Statement

All protocols for animal experiments were approved by the State Research Center for Applied Microbiology and Biotechnology Bioethics Committee (Permit No: VP-2021/4) and were performed in compliance with the NIH Animal Welfare Insurance #A5476-01 issued on 2 July 2007, and the European Union guidelines and regulations on handling, care and protection of Laboratory Animals (Available online: http://ec.europa.eu/environment/chemicals/lab_animals/home_en.htm) (accessed on 29 December 2021).

### 2.7. Animal Immunization and Exposure to Virulent Y. pestis Challenge

144 mice and 186 guinea pigs were randomly divided into six groups, and each animal was immunized subcutaneously (s.c.) with 200 μL (10^8^ CFUs) of E-BG, EYK-BG, YK-BG, SRRz-BG, or ESRRz-BG in PBS (pH 7.2) or only with PBS buffer as a placebo on days 0, and boosted on day 14. The sera of 3 mice or guinea pigs from each group were collected on 0 h and on day 28. The weight of each animal was monitored every day, until these mice and guinea pigs were sacrificed on day 28 after the first immunization. The splenic lymphocytes were isolated from these sacrificed animals.

14 d after the last immunization, mice and guinea pigs in each group were challenged s.c. with serial tenfold dilutions of two-day agar culture of wild-type *Y*. *pestis* 231 strain grown at 28 °C (six animals for a dose). The actual number of bacteria present was determined by plating on agar medium. One more group of 36 mice and 36 guinea pigs were immunized s.c. with 10 μg of F1 in 200 μL PBS (pH 7.2) adsorbed (1:10, *w/w*) to the vehicle, with an aluminum hydroxide gel colloidal suspension (Sigma-Aldrich, Denmark), or only with the PBS as a placebo and then infected with the virulent *Y. pestis* strain according to the same scheme as with the BGs preparations. Humane endpoints were strictly observed. Animals that succumbed to infection were sacrificed and examined bacteriologically to verify that infection was the cause of their death. The remaining animals were observed for 30 days.

The ability of a BG or F1 to protect an animal from death after the administration of a high dose of a virulent strain, designated as the Immunity Index (II), was calculated as the ratio: II = LD_50*imm*_/LD_50*veh*_(1)
where LD_50_ is the median lethal dose; LD_50*imm*_ is LD_50_ for animals immunized with an antigen under the study; LD_50*veh*_ is LD_50_ for vehicle-treated animals. 

### 2.8. Immune Response Assays

#### 2.8.1. ELISA

Serum IgG to *Y. pestis* BGs was measured by indirect enzyme-linked immunosorbent assay. The 96-well plates were coated with BGs (0.1 mg/mL) overnight at 4 °C. Mouse serum samples were serially diluted from 1:200 to 1:409,600, and guinea pig serum samples from 1:500 to 1:64,000. The endpoint dilution titer was calculated as the serum dilution resulting in an absorbance reading of 0.2 units above background. Goat anti-mouse IgG-HRP (Sigma, 1:5000) and goat anti-guinea pig IgG-HRP (Sigma, 1:5000) were used as the detection antibodies. The reactions were developed with TMB (3,3′,5,5′-Tetramethylbenzidine) and stopped with 2 M H_2_SO_4_. The absorbance at 450 nm was measured. Background values were obtained from serum samples collected from the animals injected with the PBS alone.

#### 2.8.2. Cellular Responses: Analysis of Stimulated Splenocytes

Cultured guinea pig and mouse splenocytes (10^6^) harvested at 14 days after the second immunization with E-BG, EYK-BG, YK-BG, SRRz-BG, ESRRz-BG, or PBS were mixed with corresponding immunogens. The cells were incubated for 48 h at 37 °C and plated in a 96-well plate coated with antibodies against IFN-γ (Guinea pig Interferon-γ ELISA Kit; CSB-E0676GU; GUSABIO; Technology LLC; Houston, TX, USA, and Mouse IFN-γ coated ELISA Kit; BMS606-2; ThermoFisher Scientific; Waltham, MA, USA) or IL- (Guinea Pig Interleukin 1β (IL-1β) ELISA Kit; CSB-E06782p GUSABIO Technology LLC, Houston, TX, USA, and Mouse IL-1 beta ELISA Kit, BMS6002; ThermoFisher Scientific; Waltham, MA, USA). ConA (5 μg/mL) (Sigma-Aldrich) at a final concentration of 5 μg/mL was used as a positive control. The standards and samples were added to the wells and the suspensions were incubated and washed with wash buffer in the kit. One hundred microliters of biotin-conjugated antibody against IFN-γ or IL-1β were used for detection. After washing, 90 μL TMB substrate was added to the wells. The suspensions were incubated for 30 min at 37 °C to visualize horseradish peroxidase activity. To stop the reaction, 50 μL stop solution was added to each well. Optical densities were measured in a microplate reader at 450 nm to calculate the IFN-γ or IL-1β concentration

### 2.9. Statistics

Specific antibodies and cytokines determination were carried out three times for the reproducibly, and the results were summarized as means ± standard error of the mean (SEM). Statistical significance was determined by *t*-test of unpaired samples and ANOVA. The results from the vaccinated groups were compared to that of the unvaccinated PBS group; statistically significant comparisons were those with *p* < 0.05. The graphs were prepared using GraphPad Prism version 8.0.0 software for Windows (GraphPad Software, San Diego, CA, USA).

The LD_50_ and a 95% confidence intervals of the virulent strains for immunized and naïve animals were calculated using the Kärber method [27].

Mortality timeframes were recorded, and the mean life to death time span was calculated for each treatment group. Comparison of the survival curves was performed using the Log-rank (Mantel-Cox) test. A *p*-value below 0.05 was considered to be significant.

## 3. Results

### 3.1. Generation and Characterization of Y. pestis Bacterial Ghosts

The λ phage holin-endolysin system is composed of S, R, and Rz/Rz1 genes, coding for holin, endolysin and accessory proteins, respectively, which are involved in bacterial cell wall degradation. Unlike the single phage ɸX174 self-sufficient lysis gene *E*, which product, independently of other genes, discontinues synthesis of the cell wall compartments locally at the site of formation of the transmembrane tunnel, the expression of endolysin is operated by holin, a small hydrophobic protein, which forms in the host cytoplasmic membrane oligomeric pores, allowing endolysin accumulated in the cytoplasm to penetrate through the pores in the inner membrane into the periplasm, where endolysin hydrolyzes all peptidoglycan available to it.

The lysis plasmids were transformed into *Y. pestis* KM260(12)Δ*lpxM* strain to obtain BGs. The OD_600_ values of the E-BG, EYK-BG, YK-BG, SRRz-BG, and ESRRz-BG suspensions were determined at different incubation time, and the growth curves of E-BG, EYK-BG, YK-BG, SRRz-BG, and ESRRz-BG were prepared based on the measured OD_600_ values at each time point (Figure 2A). The OD_600_ of E-BG, EYK-BG, YK-BG, SRRz-BG, and ESRRz-BG was reduced constantly after a shift in temperature. Cell viability also decreased after lysis induction (Figure 2B). Derivatives of the parent strain carrying different lysis plasmids lost their viability after induction of phage genes at different rates for 4–8 h (Figure 2B), with the exception of E-BG, which lost its viability only by 24 h (data not shown).The number of YK-BG viable cells declined faster than those of SRRz-BG, ESRRz-BG and EYK-BG, as evidenced by the lower number of CFU observed in ghost preparations (Figure 2B). In contrast, *Y. pestis* KM260(12)Δ*lpxM*/pEYR’carrying the empty cloning vector was not hampered in its growth behavior at 42 °C for up to 8 h and showed only a slight reduction after 24 h (data not shown). The lysis rate of induced mutant BGs was counted as 99.99% ± 0.01% when the YP BGs were harvested 24 h after the induction (data not shown).

The formation of *Y. pestis* ghosts and release of cell contents were confirmed under the transmission electron microscope (Figure 3B–F) by comparing with the *Y. pestis* KM260(12)Δ*lpxM*/pEYR’ carrying the empty cloning vector (Figure 3A). Ninety eight percent of bacteria in the population of the precursor strain had a fine structure typical of gram-negative bacteria. The components of the cell wall are clearly visible only on a part of the cell surfaces in about half of the bacteria that do not produce phage proteins (Figure 3A). The outer membrane is thin. The subtle periplasm is filled with a relatively electron-dense peptidoglycan. The cytoplasmic membranes are relatively smooth. The cytoplasm is filled with various globular and fibrillar components that form an intense diffuse uneven electron density. The nucleoids are localized in the electron-transparent zones of the cytoplasm. The remaining less than 2% of bacteria had irreversible structural damages to the cell walls, cytoplasm and nucleoids.

Electron microscopic analysis did not reveal the formation of lysis pores in *Y. pestis* ghosts (Figure 3B–F), but there were other signs of an impaired integrity of the bacterial cell wall. A TEM evaluation showed that *Y. pestis* ghost cells were empty due to the loss of cytoplasmic material and had collapsed cell envelopes compared to the cells that were free from phage lysine genes (Figure 3A). Expression of the E (Figure 3B) protein is accompanied by increase of transparency of periplasm, which provides visualization of the outer membranes and the outer border of the cytoplasmic membranes.

Bacterial cells practically retain their shape, but the cell wall loses its roundness and becomes wrinkled in places. The thickness of the periplasm does not increase, but becomes free from the electron-dense matter. The cytoplasm becomes less electron-dense, which is accompanied by the appearance of clear boundaries in the electron-dense granules and bodies of intracellular contents. Only 3% of E-BG^+^ lacked irreversible structural damage.

Micrographs of bacteria producing holin and endolysin of the phage L-413C practically do not differ, regardless of the presence (Figure 3E) or absence (Figure 3C) of combined protein E synthesis. Bacterial cells completely lose their three-dimensional forms, resembling not “bacterial ghosts”, but rather crumpled cloth bags, which led us to call this variant “bacterial shreds”.

The presence in the strain of the genes of holin and endolysins of the phage λ (Figure 3D,F) is accompanied by the tendency of bacterial ghosts to take a more rounded shape than in classical E-induced BGs with even more wrinkled cell walls. In separate fields of view, BGs are visible that are not closed in a ring. Against the background of the general clearing of the BGs content, in about a third of the cells, accumulations of electron-dense material with diffuse (Figure 3D) or clear boundaries (Figure 3F) are visible. The strain carrying the genes of holin and endolysin of phage λ in combination with the gene for protein E (Figure 3F), in addition to BGs with clearly limited cytoplasmic membranes with very electron-dense contents, form “bacterial shreds”, similar to those induced by the action of holin and endolysin of the phage L-413C (Figure 3C,E).

### 3.2. Humoral Immune Responses

YP BGs specific IgG antibody titers in mice and guinea pigs’ sera were monitored during immunization of the six treatment groups. As shown in Figure 4A,B, no specific IgG antibodies were detected in the PBS treated animals. E-BG, EYK-BG, YK-BG, SRRz-BG, and ESRRz-BG-s.c. inoculation primed similar levels of total anti-BG IgG in mice without substantial difference (*p* > 0.05). The highest IgG titer in guinea pigs induced by SRRz-BG, EYK-BG, and ESRRz-BG was 32,000, 16,000 and 16,000, respectively *(p* < 0.001) (Figure 4). The titers of the E-BG and YK-BG guinea pigs groups were lower than that of SRRz-BG, EYK-BG, and ESRRz-BG groups (*p* < 0.001).

### 3.3. IFN-γ Analysis

To further understand the immune mechanisms induced by BGs, the expression of cytokine IFN-γ for guinea pigs and mice was assessed. As shown in Figure 5, the PBS-treated groups showed an inability to induce IFN-γ production. In contrast, significant increases in the IFN-γ levels were detected in the YK-BG and especially in the EYK-BG-treated groups of guinea pigs following the last immunization (*p* < 0.05). The ability of BGs to induce IFN-γ production in guinea pigs was noticeably higher.

### 3.4. Analysis of Inflammatory Response

Equally high levels of IL-1β, were observed in mice treated with any variant of BGs (*p* < 0.05). In all other collected samples the levels of IL-1β were at least two times lower (*p* < 0.05) than that induced by BGs in mice. No cytokine increase was detected in animal groups treated by PBS (*p* < 0.01).

### 3.5. Protection of Immunized Animals against Y. pestis Challenge

Indices of immunity induced by BGs variants, the ratio of LD_50_ values in immunized animals to similar indicators in the naïve ones, were determined for a comparative assessment of the protective potency in relation to the two species of laboratory animals (Table 3). The murine immune system practically did not induce a protective response to the inoculation of BGs, while guinea pigs, on the contrary, formed a significant protective immunity, especially against YK-BG and EYK-BG. F1, on the other hand, provided reliable protection for mice, but did not affect the survival of guinea pigs (Figure 6). No abnormal behaviors were observed in the immunized rodents, and no obvious difference of weight was detected between the animals from different groups.

## 4. Discussion

Plague is a re-emerging zoonotic bacterial infection whose causative agent, *Y. pestis*, is circulating in natural plague foci covering approximately 6–7% of the land area and occasionally causing pandemics that have claimed hundreds of millions of lives [28]. Live and killed plague vaccines of the first generation helped to stop the third pandemic [14]. In several countries, live vaccine based on the *Y. pestis* EV76 vaccine strain is still used for epidemic indications, although it has long been required to be replaced with medications that meet modern requirements [29]. Modern plague vaccines candidates are being developed by a number of laboratories around the world using various state-of-the-art technologies. As a rule, the majority of them are developed on the basis of the two antigens that are highly protective for mice, F1 and V, which protect other animal species to a lesser extent [13]. The F1-V fusion protein vaccine protected cynomolgus macaques, but largely failed to protect African green monkeys [30], raising concerns that humoral immunity targeting F1 and V might be inadequate in protecting people from pneumonic plague. According to Li et al., in the case of post-infection immunity, the seroprevalence to the F1 antigen in all patients recovered from plague was 78.5% [31], while in the case of the V antigen it was only 28.6% [32].

Also, LcrV was shown to be a poor marker of successful immunization with live plague vaccine [33]. The outcomes of bacterial infections and the immune response to bacterial vaccines are highly variable in different animal species and even in diverse intraspecies groups [34,35,36]. At least a partially successful solution to this problem is generated by replacing very popular inbreds of one mammalian species with outbred biomodels of two or three species [2,37,38], or with carefully selected panels of several inbreds [39]. Both of these approaches aim to reproduce the diversity of clinical pictures and immune responses detected in the natural animal and human populations. However, this approach will only provide an understanding of what proportion of the immunized remained unprotected, but in no way will protect seronegatives and/or T cell-negatives. To ensure the protection of that part of the immunized that does not respond to the antigens already included in the vaccine, it is necessary to introduce into the vaccine composition of conservative protective antigens, characterized by minimal polymorphism of their molecular structure and epitopes, as well as the uniformity of the protective immune response in different genotypes of immunized individuals.

In our preliminary research, we faced a similar problem. Our team also developed such an F1 and V two-component vaccine and received a national license (registration certificate) for the production of this molecular microencapsulated plague vaccine [40]. However, 33% of the human subjects did not develop specific antibodies to either the V or F1 antigen during 90 days of observation post vaccination. It has been shown that different species and even intraspecies groups of mammals react differently to the same *Y. pestis* antigens [41] and strains [42]. Although today there is no consensus on the *Y.-pestis*-susceptible biomodel that is closest to the plague in humans, most researchers chose mice. This choice is based on low cost, small size, and availability of facilities to perform containment work [43]. The guinea pig is another small animal model. These animals are several times more expensive than mice, in contrast to which their serum, like human serum, is bactericidal for some strains of *Y. pestis* belonging to the subsp. *microti* bv. Caucasica. Strains of the *microti* subspecies are highly virulent for mice and are only relatively pathogenic for guinea pigs and humans [44]. Cavies (guinea pigs), like humans, are not susceptible to the murine toxin [45]. The closeness of the guinea pigs to humans in a variety of responses to *Y. pestis* and its components suggest that antigens that are protective for cavies are more likely to be protective for humans than protective for mice.

A variety of antigens other than F1 or V have been evaluated for inclusion into the plague vaccine [11], but they were tested in a mouse model that has its aforementioned limitations. The possibility of the formation of specific immunity to plague in guinea pigs is associated mainly with still unidentified water-insoluble components of the microbe [5,6]. There is certain diversity among humans and laboratory animals in innate and adaptive anti-infection immunity [46]. These differences including the host-species-specific differences in interferon γ- and IL-1β-mediated effector mechanisms are still little studied. We also failed to make global discoveries in this section of our research. However, we showed that the BG-induced increase in the levels of IL-1β in the blood of the both guinea pigs and mice does not correlate with the better protection, and in the case of an increase in the level of interferon γ, a correlation with the degree of protection was observed only in guinea pigs. To close gaps in our knowledge of human plague immunogenesis, future investigations with the use of humanized animal models should be carried out.

The main goal of our current study was to determine the presence in BGs preparations of components that protect guinea pigs (and may be humans), and the dependence of the potency of the immune response on the degree of degradation of the BGs cell wall.

At the first stage of the work, it was planned to develop preparations of *Y. pestis* bacterial ghosts lysed to varying degrees [47] and to carry out a comparative assessment of their protective potency for mice and guinea pigs.

Taking into account the possibility of the presence of viable bacteria in BGs preparations obtained using both the protein-E-mediated lysis and the holin-endolysin system [21], we chose as the basis for vaccine candidates the avirulent *Y. pestis* KM260(12) strain, devoid of the three resident plasmids. In addition to attenuation, the loss of these plasmids was accompanied by the loss of ability to produce a number of proteins encoded by them, including harmful factors such as plasminogen activator (Pla) and “murine” toxin [48], as well as a number of non-protective ballast antigens, such as Pst, YopE, etc. [5]. The loss of the two main protective antigens F1 and LcrV encoded by these plasmids was planned to be compensated by introducing purified recombinant proteins into vaccine candidates in our subsequent studies.

The approach we have chosen to construct BGs turned out to be correct and, judging by the TEM data, made it possible to generate an almost complete spectrum of structural variants of BGs determined by the degree of degradation of the peptidoglycan skeleton. A comparison of TEM photographs of the *Y. pestis* cell surface after lysis induction showed that BGs harboring gene *E* [21] or genes *S*, *R* and *Rz* [49] showed noticeable surface folds (loss of roundness) from the loss of cytoplasmic contents, though retained the basic morphology of the parent bacterial cells. Relatively complete outer membranes with small changes in morphology indicate only local hydrolysis of peptidoglycan. It is obvious that protein E hydrolyzes the murein layer only at the site of the formation of the transmembrane tunnel [21], and Sam7 mutation of the λ phage lysis genes complicates the transition of endolysin from the cytoplasm to the periplasm [50] and, accordingly, the hydrolysis of peptidoglycan. We did not consider the dynamics of changes in cell morphology of individual strains of *Y. pestis*, but, most likely, derivatives expressing protein E or the “semi-functional” holin-endolysin system of λ phage, not even within 24, but 48 h after induction, will not display a picture similar to that characteristic of the Y^+^ K^+^ variant with the holin-endolysin functional system of the L-413C phage, which is responsible for the complete annihilation of murein in the bacterial suspension. At the same time, the tunnel-forming protein E causes local hydrolysis of peptidoglycan only at the site of the transmembrane tunnel, and the holin system of λ phage with a mutant holin gene can hydrolyze peptidoglycan only in part of the cells of the strain carrying such a partially functional lytic plasmid. Increasing the duration of action is unlikely to change the molecular target.

One of the main goals of introducing bacterial ghost technology is to create structures with minimally altered bacterial surfaces. It was expected that better preserved cell wall preparations would induce better protection. However, both in our studies and in the work of others [21], a significant increase in protection was noted in variants of BGs with a completely destroyed peptidoglycan skeleton. In comparison with the PBS injection, immunization with the E-Y-K-BGs protected guinea pigs against 333 LD_50_ of *Y. pestis* infection.

It has been shown that peptidoglycan has both immunosuppressive [51] and immunostimulating properties [52]. Hydrolysis of peptidoglycan leads to the formation of many muramyl-dipeptide (*N*-acetyl-muramyl-L-alanyl-D-isoglutamine) molecules, the smallest adjuvant active moieties capable of replacing whole killed mycobacterial cells in Freund’s complete adjuvant [53] and, accordingly, of increasing the immune response.

## 5. Conclusions

In summary, our E-Y-K-BGs vaccine candidates with fully hydrolyzed peptidoglycan induce significantly greater protection from *Y. pestis* in guinea pigs in comparison with classical BGs and can be used as a relatively effective vector for the development of a safe plague vaccine for multiple hosts (human, mouse, and cattle).

Regardless of whether our hypothesis about a peptidoglycan-dependent BGs mechanism of increasing protective potency for guinea pigs is true or not, we managed to generate a candidate plague vaccine that induces intense immunity in guinea pigs (immunity index = 49,000) comparable to that developing in response to immunization with live plague vaccine (immunity index = 9,400,000 [13]). In subsequent experiments, we plan to evaluate how much the protective potency of our BGs can be increased for guinea pigs when they are co-administered with F1 and V antigens as well as the effectiveness of such a three-component vaccine candidate for mice.

Our data clearly suggest that selection of protective antigens for multiple host species vaccines should be carried out from the very beginning using as many animal species that this vaccine should protect as possible.

## Figures and Tables

**Figure 1 vaccines-10-00051-f001:**
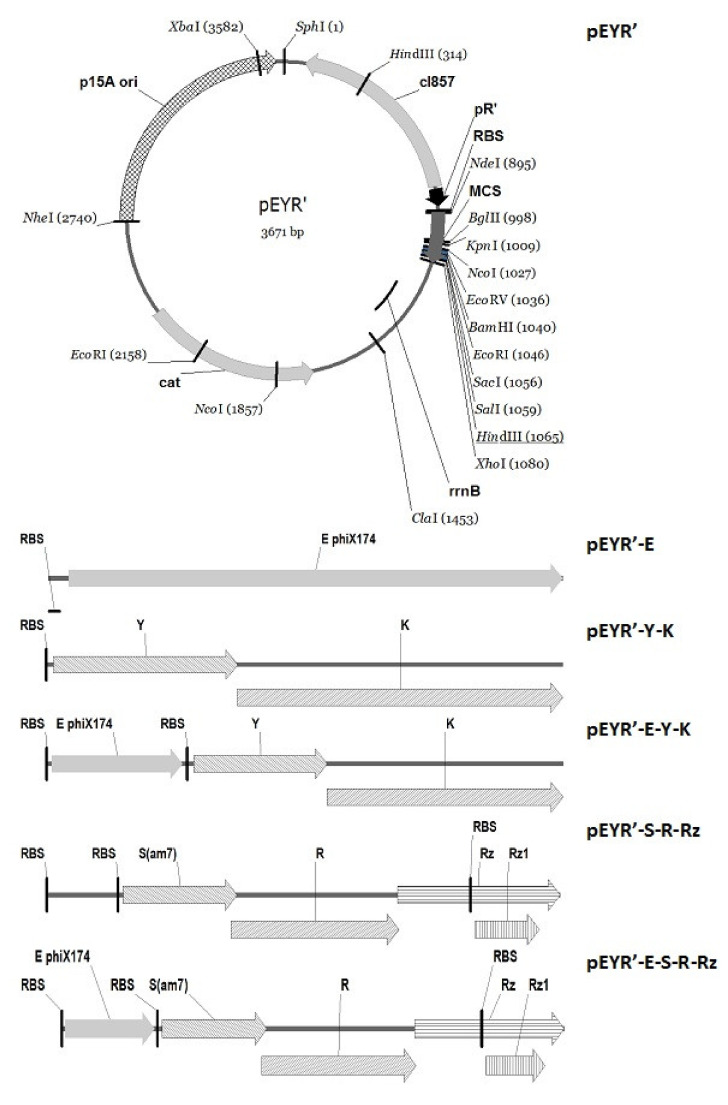
Schematic representation of vector and lysis cassettes used in this study.

**Figure 2 vaccines-10-00051-f002:**
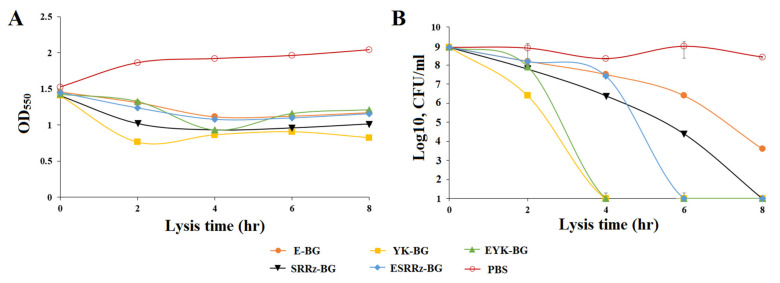
The lysis kinetics of *Y. pestis* KM260(12)ΔlpxM harboring different lysis plasmids. Lysis was monitored by the measurement of the OD_600_ (**A**) and the determination of the number of CFU (**B**). *Y. pestis* BGs grown to exponential phase were inactivated by induction of the lysis genes. The CFU counts were transformed to log base 10 values. The data are presented as the mean ± s.d. of three samples.

**Figure 3 vaccines-10-00051-f003:**
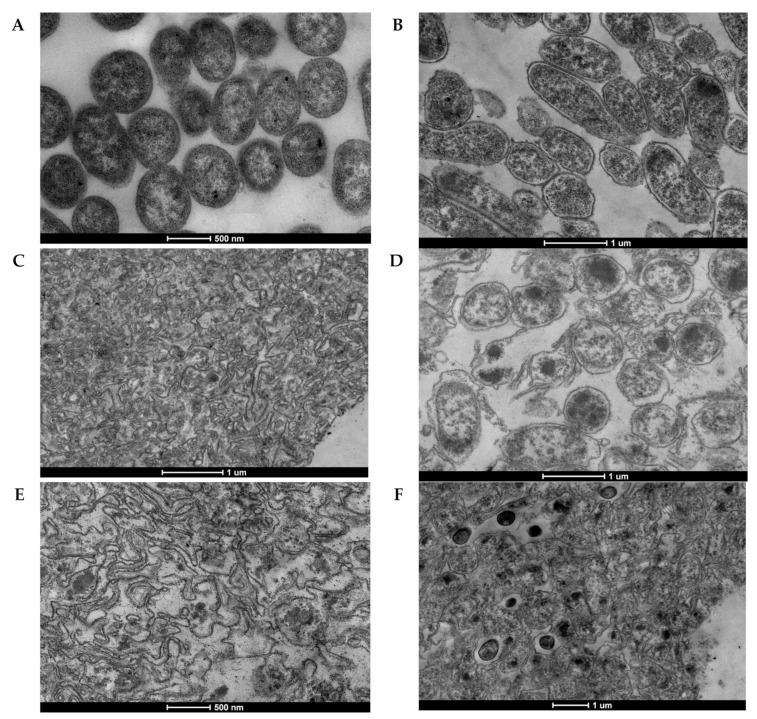
Transmission electron micrographs of *Y. pestis* strains (**A**) KM260(12)∆lpxM, (**B**) KM260(12)∆lpxM) pEYR’-E, (**C**) KM260(12)∆lpxM/pEYR’-Y-K, (**D**) KM260(12)∆lpxM/pEYR’-S-R-Rz KM260(12)∆lpxM/pEYR’-S-R-Rz, (**E**) KM260(12)∆lpxM/pEYR’E-Y-K, (**F**) KM260(12)∆lpxM pEYR’-E-S-R-Rz. (24 h after the induction).The bar represents 0.1 μm (**B**–**D**,**F**) or 500 nm (**A**,**E**).

**Figure 4 vaccines-10-00051-f004:**
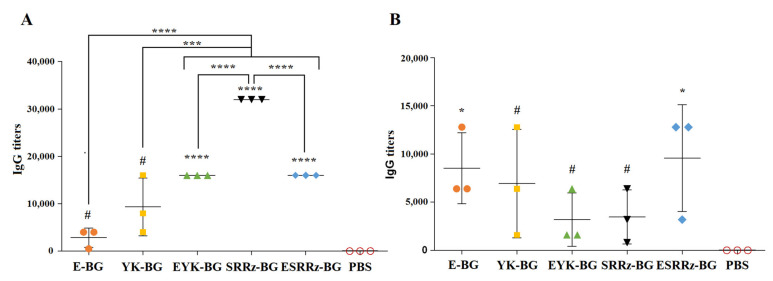
Antibody response in sera of guinea pigs (**A**) and mice (**B**) immunized s.c. with E-BG, EYK-BG, YK-BG, SRRz-BG, ESRRz-BG, and PBS, respectively at week 4 post administration. 100 μL of *Y. pestis* BGs were coated as antigen, the goat anti-mouse IgG-HRP or goat anti-guinea pig IgG-HRP was used as detection antibody. ^#^—*p* > 0.05; *—*p* < 0.05; ***—*p* < 0.001; ****—*p* < 0.0001.

**Figure 5 vaccines-10-00051-f005:**
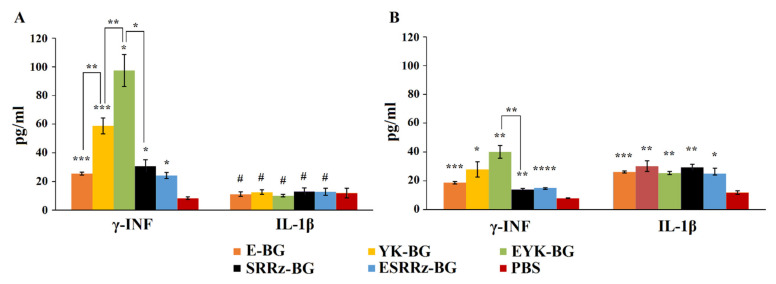
Cytokines levels of splenic lymphocytes from immunized guinea pigs (**A**) and mice (**B**). The splenic lymphocytes were isolated on day 14 after the last immunization, and corresponding BGs were used as immunogens. The culture supernatants were harvested after 48 h and the cytokine concentration was measured by ELISA. ^#^—*p* > 0.05; *—*p* < 0.05; **—*p* < 0.005; ***—*p* < 0.001. ****—*p* < 0.0001.

**Figure 6 vaccines-10-00051-f006:**
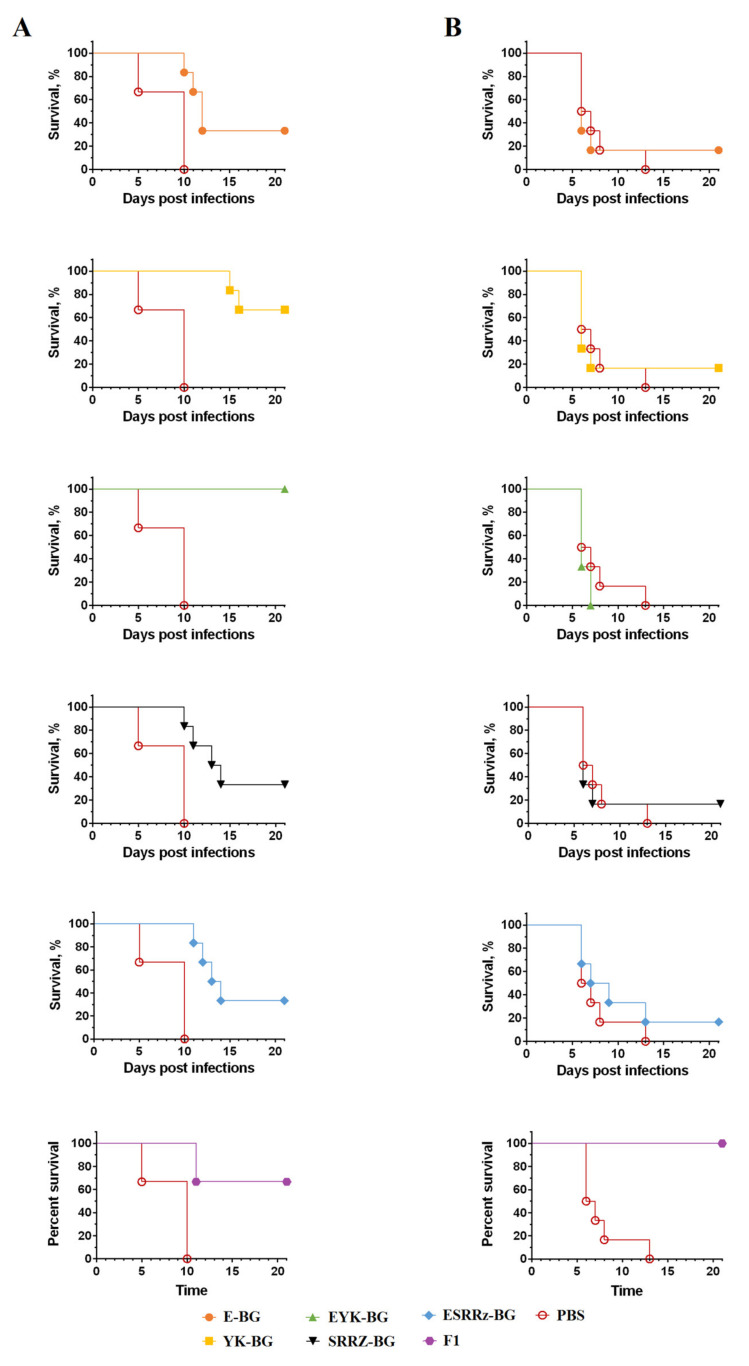
Protection of YP BGs against a lethal challenge with the wild-type *Y. pestis* 231 strain. Guinea pigs (**A**) and mice (**B**) were subjected to s.c. infection with different BGs strains or F1 at day 0 and boosted at day 14. 14 days after the last immunization, six mice and six guinea pigs from each group were challenged s.c. with 10^2^ CFUs of wild-type *Y. pestis* 231 (33.3 LD_50_ for mice and 14.3 LD_50_ for cavies).

**Table 1 vaccines-10-00051-t001:** Bacterial strains, plasmids, and bacteriophages used in this study.

Strains/Plasmid/Bacteriophage	Relevant Genotype or Annotation	Source
*E. coli*		
DH5α	F^−^, *gyrA96*(Nal^r^), *recA1*, *relA1*, *endA1*, *thi-1*, *hsdR17*(r_k_^−^, m_k_^+^), *glnV44*, *deoR*, Δ(*lacZYA*-*argF*)*U169*, [φ80dΔ(*lacZ*)*M15*], *supE44*	SCPM-O *
*Y. pestis*		
231	wild type strain; universally virulent (LD_50_ for mice ≤ 10 CFU, for guinea pigs ≤ 10 CFU); Pgm^+^, pMT1, pPCP1, pCD^−^	SCPM-O [23]
KM260(12)	avirulent derivative of 231; Pgm^+^, pMT1^−^, pPCP1^−^, pCD^−^	SCPM-O
KM260(12)Δ*lpxM*	Δ*lpxM* derivative of KM260(12)	SCPM-O
KM260(12)Δ*lpxM*/pEYR’	KM260(12)Δ*lpxM* containing pEYR’	This study
KM260(12)Δ*lpxM*/pEYR’-E	KM260(12)Δ*lpxM* containing pEYR’-E	This study
KM260(12)Δ*lpxM*/pEYR’-E-Y-K	KM260(12)Δ*lpxM* containing pEYR’-E-Y-K	This study
KM260(12)Δ*lpxM*/pEYR’-Y-K	KM260(12)Δ*lpxM* containing pEYR’-Y-K	This study
KM260(12)Δ*lpxM*/pEYR’-E-S-R-Rz	KM260(12)Δ*lpxM* containing pEYR’-E-S-R-Rz	This study
KM260(12)Δ*lpxM*/pEYR’-S-R-Rz	KM260(12)Δ*lpxM* containing pEYR’-S-R-Rz	This study
Plasmid		
pACYC184	Source of p15A *ori* and *cat* gene	[24]
pBAD/myc-HisA	Source of *rrnB* transcription terminator	Invitrogen
pET32b (+)	Source of multiple cloning site	Novagene
pEYR’	Expression vector, phage Lambda modified right promoter (pR’) (Cm^r^)	SCPM-O
pEYR’-E	Lysis plasmid, pEYR’-lysis E (Cm^r^)	SCPM-O
pEYR’-E-Y-K	Lysis plasmid, pEYR’-lysis E, Y, K (Cm^r^)	SCPM-O
pEYR’-Y-K	Lysis plasmid, pEYR’-lysis Y, K (Cm^r^)	SCPM-O
pEYR’-E-S-R-Rz	Lysis plasmid, pEYR’-lysis E, S, R, Rz (Cm^r^)	SCPM-O
pEYR’-S-R-Rz	Lysis plasmid, pEYR’-lysis S, R, Rz (Cm^r^)	SCPM-O
Bacteriophage		
λCE6	Source of pR promoter, holin (*S*) and endolysins (*R-Rz*) genes (*cI857Sam7*)	Thermo Scientific
φX174	Source of E protein gene	Thermo Scientific
L-413C	Source of holin (*Y*) and endolysin (*K*) genes	Russian Research Anti-Plague Institute Microbe

* The State Collection of Pathogenic Microbes and Cell Cultures on the base of the State Research Center for Applied Microbiology and Biotechnology (“SCPM-Obolensk”).

**Table 2 vaccines-10-00051-t002:** Primers used in this study.

Name	Sequence
pEY-1 (ClaI)	5′CCCATCGATCATATCGTCAATTATTAC3′ *
pEY-2 (SphI)	5′ATATTGCATGCTGTCAAACATGAGAATTAC3′
pEY-5 (NdeI)	5′GGCCATATGCACCATCATCATC3′
pEY-6^б^	5′*TCAGTGGTGGTGGTGGTGGTG*3′ ****
pEY-7^б^	5′*CACCACCACCACCACCACTGA*TGAGTTTAAACGGTCTCCAG3′
pEY-8 (ClaI)	5′CCCATCGATTTGCTTCGCAACGTTCAAATC3′
pR-For (SphI)	5′CACAAAGCATGCGGAGTGAAAATTCCCCTAATTCG3′
pR-Rev (NdeI)	5′GATACCATATGAACCTCCTTAGTACATGCAACCATT3′
pR (T > C) (NdeI)	5′GTGCATATGAACCTCCTTAGTACATGCAACCATTATCACCGCCAGAGGTAAAATAGTCAACACGCGCGGTGTTAG3′
E1 (NdeI)	5′AGGCATATGGTACGCTGGACTTTGTG3′
E2 (XhoI)	5′AATCTCGAGTCACTCCTTCCGCACGTAA3′
Y-NdeI (XhoI)	5′GGTGGCATATGACAGCAGAAGAAAAAAGC3′
Y-SalI (SalI)	5′GCCGTCGACAACAGGAGGAATTAACCATGACAGCAGAAGAAAAAAGC3′
K-XhoI (XhoI)	5′ATTCTCGAGTTAAGCCGGTACGCCGCCAG3′
S-R-Rz-For (SalI)	5′AAAGTCGACAACAGGAGGAATTAACCATGAAGATGCCAGAAAAACATG3′
S-R-Rz-Rev (XhoI)	5′ATTCTCGAGCTATCTGCACTGCTCATTAAT3′

* Restriction sites are underlined, ** Complementary sequences are highlighted in italics.

**Table 3 vaccines-10-00051-t003:** Indices of immunity (II) induced by BGs variants.

Animals Immunized with BGs/Antigen	Guinea Pigs		Mice	
	LD_50_, CFU *	II **	LD_50_, CFU	II
….	147 37 ÷ 584	4.9 × 10^1^	32 8 ÷ 126	4.6 × 10^0^
Y-K	6813 2154 ÷ 27123	2.3 × 10^3^	22 5 ÷ 86	3 × 10^0^
E-Y-K	146,780 36,869 ÷ 926,119	4.9 × 10^4^	22 5 ÷ 86	3 × 10^0^
S-R-Rz	68 17 ÷ 271	2.3 × 10^1^	15 4 ÷ 58	2 × 10^0^
E-S-R-Rz	147 37 ÷ 584	4.9 × 10^1^	46 12 ÷ 233	6.6 × 10^0^
F1 **	316 79 ÷ 1259	1.0 × 10^2^	100,000 25,119 ÷ 630,957	1.4 × 10^4^
PBS	3 1 ÷ 13	1 × 10^0^	7 2 ÷ 27	1 × 10^0^

* Values are given as means ±95% confidence intervals. ** The ability of BGs to protect an animal from death after administration of a high dose of a virulent wild type strain, designated Immunity Index (II) was calculated as the ratio: II = LD_50*imm*_/LD_50*veh*_.

## Data Availability

All the data will be provided on reasonable request.

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
