# Peer review of "Peptidoglycan-Free Bacterial Ghosts Confer Enhanced Protection against Yersinia pestis Infection"

_vaccines, 2021, doi:10.3390/vaccines10010051_

Round 1

Reviewer 1 Report

This is an interesting study evaluating the utility of bacterial ghosts in protecting against Yersinia pestis infection. The manuscript is well-written, and study design is appropriate. Data is very clearly presented. The manuscript is particularly interesting as in addition to evaluating bacterial ghosts as vaccine candidates, the authors evaluate different "levels" of lysis by the system, as well as two different animal models. Overall it is a nice study with somewhat compelling results. In some cases, the manuscript can use some clarification for the purpose of presentation. Minor comments/suggestions follow:

It might be useful to explicitly state up front (either at the end of the introduction or in the beginning of the results) that the purpose of the different constructs was to generate different degrees of lysis for evaluation. This becomes clear in the discussion, but would be helpful early. (if this is not the case and this wasn't the purpose, obviously disregard this suggestion)

 In results section 3.1, a bit of detail (brief) about what genes the lysis plasmids carry, the resulting proteins, and how they work would be helpful. This information is elsewhere in the manuscript, but would be helpful here. In short, a bit of an introduction to the system for generating ghosts. Further, setup for why the different constructs are being used (different degrees of lysis?) would be useful.

Figure 3 is very impressive,  but could use some slight clarification. Please specify if this is taken from a specific timepoints that could be compared to the data seen in Figure 2. There are a range of results here- is one preferred, and why? Also, obviously C, E, and F look very much like killed whole cell bacterial lysates. Perhaps in the discussion it may be worth differentiating lysates from what we see in the figure if BG are preferred. why would highly lysed BG be preferential as a vaccine candidate compared to whole cell lysates?

For figures including mice and guinea pigs, please specify (A) or (B) which are showing guinea pigs and which are showing mice in the figure legend. 

For section 3.3, please indicate why IFN-g and IL-1B are being tested, and are the only two cytokines tested. Are increased levels of either/both correlated with better protection?  

Author Response

Author's Reply to the Reviewer’s Comments

Reviewer 1

This is an interesting study evaluating the utility of bacterial ghosts in protecting against Yersinia pestis infection. The manuscript is well-written, and study design is appropriate. Data is very clearly presented. The manuscript is particularly interesting as in addition to evaluating bacterial ghosts as vaccine candidates, the authors evaluate different "levels" of lysis by the system, as well as two different animal models. Overall, it is a nice study with somewhat compelling results. In some cases, the manuscript can use some clarification for the purpose of presentation.

RESPONSE: Thank you for the appreciation of our work.

Minor comments/suggestions follow:

It might be useful to explicitly state up front (either at the end of the introduction or in the beginning of the results) that the purpose of the different constructs was to generate different degrees of lysis for evaluation. This becomes clear in the discussion, but would be helpful early. (if this is not the case and this wasn't the purpose, obviously disregard this suggestion)

RESPONSE: According to your recommendation, the clarification has been added at the end of the introduction section:

“The purpose of the design of different constructs was to generate diverse degrees of lysis for evaluation of the interrelation of the degree of degradation of the bacterial cell wall with the intensity of immunity to plague infection in mice and guinea pigs”.

 In results section 3.1, a bit of detail (brief) about what genes the lysis plasmids carry, the resulting proteins, and how they work would be helpful. This information is elsewhere in the manuscript, but would be helpful here. In short, a bit of an introduction to the system for generating ghosts. Further, setup for why the different constructs are being used (different degrees of lysis?) would be useful.

RESPONSE: The explanation is added.

Figure 3 is very impressive, but could use some slight clarification. Please specify if this is taken from a specific timepoints that could be compared to the data seen in Figure 2. There are a range of results here- is one preferred, and why? Also, obviously C, E, and F look very much like killed whole cell bacterial lysates. Perhaps in the discussion it may be worth differentiating lysates from what we see in the figure if BG are preferred. why would highly lysed BG be preferential as a vaccine candidate compared to whole cell lysates?

RESPONSE: We tried to describe these manipulations more clearly (see lines 131-134, 144 and 282, 443-453). Unfortunately, we did not look at the dynamics of changes in the morphology of individual strains, but we do not think that if we studied the derivative expressing protein E not 24 but 48 hours after induction, then the picture would be similar to that of the variant with the holin-endolysin system of the plague diagnostic phage. As noted on line 439, the tunnel-forming protein E causes only local hydrolysis of peptidoglycan at the site of the transmembrane tunnel, while intact holin-endolysin system is able to hydrolyze all cellular peptidoglycan. Protein E does not hydrolyze all of the murein within a single cell, while phage λ holin-endolysin system with a defective holin gene does not hydrolyze peptidoglycan in all the cells of the strain harboring such partially functional lytic plasmid (lines 440-443). These clarifications are included in the revised manuscript.

For figures including mice and guinea pigs, please specify (A) or (B) which are showing guinea pigs and which are showing mice in the figure legend. 

RESPONSE: The omission has been corrected.

For section 3.3, please indicate why IFN-g and IL-1B are being tested, and are the only two cytokines tested.

RESPONSE: The use of only two cytokines is not our fault, but our misfortune. Scientists in Russia are experiencing serious problems with obtaining reagents. For example, the delivery time for "PINK BENGAL", in the absence of stock, will be from 90 to 180 working days (https://nevareaktiv.ru›qa). Many reagents, which are not produced in Russia, are not available at all. As a result, from a whole list of test kits for determining the content of cytokines in the blood of mice and guinea pigs, we managed to order only those diagnostics that were used in our work. However, it seems to us pas comme il faut to write about this reason in a scientific paper.

Are increased levels of either/both correlated with better protection?  

RESPONSE: Thank you for pointing this out. Information on correlation on cytokine levels with better protection was added on lines 412-417.

ded into the legend of Figure 6.

Reviewer 2 Report

The authors used  hypo-endotoxic Yersinia pestis bacterial 17 ghosts (BGs) with combinations of genes encoding the bacteriophage ɸX174 lysis-mediating protein 18 E and/or holin-endolysin systems from λ or L-413C phages to develop a modern plague vaccine. The research design and methods are appropriate, results are clearly represented and conclusions are well supported by the results. Therefore, I believe this paper is worth enough to be published in the journal vaccines.  

Author Response

Author's Reply to the Reviewer’s Comments

Reviewer 2

The authors used  hypo-endotoxic Yersinia pestis bacterial 17 ghosts (BGs) with combinations of genes encoding the bacteriophage ɸX174 lysis-mediating protein 18 E and/or holin-endolysin systems from λ or L-413C phages to develop a modern plague vaccine. The research design and methods are appropriate, results are clearly represented and conclusions are well supported by the results. Therefore, I believe this paper is worth enough to be published in the journal vaccines.  

RESPONSE: Thank you for reading and appreciating our manuscript.

Reviewer 3 Report

In this study, the authors generated peptidoglycan-free bacterial ghosts using different lysis proteins encoded by different phage species. Their research objective was to develop a better vaccine for Yersinia pestis, as current live, inactivated, and subunit vaccines have limitations. Bacterial ghosts can potentially address this issue , and their ability to generate a protective immune response was measured. The researchers found that the E-Y-K BGs induced significantly greater protection in guinea pigs but not in mice. Further studies are needed to continue to assess the potential of these BGs to form an effective vaccine.  Overall, I found this study to be well designed and written with sufficient statistical analysis done. I have some comments below that I believe will improve the quality of the published manuscript but I believe it is suitable for publication in its current form. 

Why did you focus on just these two cytokines? no explanation is given. 

Table 1: I believe the source is misspelled as sours

Figure 2/4/5/6 

Several issues can be adjusted to improve clarity.

Use a similar color pattern or shape throughout these figures, don't switch between the two. 

For figure 4 & 5 use an identical figure type to improve clarity.

Either label each panel as mouse/ guinea pig or provide text in the legend, in its current form it is not clear which panel belongs to each species.

Either use PBS or control, don't switch between the two.

Figure 4 IgG titer is misspelled

Figure 6. Guinea pigs in panel A and Mice in panel B? Clearly label the two panels.

Author Response

Author's Reply to the Reviewer’s Comments

Reviewer 3

In this study, the authors generated peptidoglycan-free bacterial ghosts using different lysis proteins encoded by different phage species. Their research objective was to develop a better vaccine for Yersinia pestis, as current live, inactivated, and subunit vaccines have limitations. Bacterial ghosts can potentially address this issue, and their ability to generate a protective immune response was measured. The researchers found that the E-Y-K BGs induced significantly greater protection in guinea pigs but not in mice. Further studies are needed to continue to assess the potential of these BGs to form an effective vaccine. Overall, I found this study to be well designed and written with sufficient statistical analysis done. I have some comments below that I believe will improve the quality of the published manuscript but I believe it is suitable for publication in its current form. 

RESPONSE: Thank you for your scrupulous reading of our manuscript and helpful comments.

Why did you focus on just these two cytokines? no explanation is given. 

RESPONSE: The use of only two cytokines is not our fault, but our misfortune. Scientists in Russia are experiencing serious problems with obtaining reagents. For example, the delivery time for "PINK BENGAL", in the absence of stock, will be from 90 to 180 working days (https://nevareaktiv.ru›qa). Many not produced in Russia reagents are not available at all. So, from a whole list of test kits for determining the content of cytokines in the blood of mice and guinea pigs, we managed to order only those diagnostics that were used in our work. However, it seems to me pas comme il faut to write about this reason in a scientific paper.

Table 1: I believe the source is misspelled as sours

RESPONSE: Thank you. The mistake is corrected.

Figure 2/4/5/6 

Several issues can be adjusted to improve clarity.

Use a similar color pattern or shape throughout these figures, don't switch between the two. 

RESPONSE: Thank you for pointing this out. The proposed amendments were made.

For figure 4 & 5 use an identical figure type to improve clarity.

Either label each panel as mouse/ guinea pig or provide text in the legend, in its current form it is not clear which panel belongs to each species.

RESPONSE: The required addition is included into the legend of Figures 5 & 6.

Either use PBS or control, don't switch between the two.

RESPONSE: We chose PBS.

Figure 4 IgG titer is misspelled

RESPONSE: Thank you. The mistake is corrected.

Figure 6. Guinea pigs in panel A and Mice in panel B? Clearly label the two panels.

RESPONSE: The required addition is included.
